# Rates of obstetric intervention and associated perinatal mortality and morbidity among low-risk women giving birth in private and public hospitals in NSW (2000–2008): a linked data population-based cohort study

Hannah G Dahlen,[1] Sally Tracy,[2] Mark Tracy,[3,4] Andrew Bisits,[5,6] Chris Brown,[7] Charlene Thornton[1]

For numbered affiliations see end of article.

Correspondence to
Professor Hannah G Dahlen;
h.dahlen@uws.edu.au

## ABSTRACT

**Objectives:** To examine the rates of obstetric intervention and associated perinatal mortality and morbidity in the first 28 days among low-risk women giving birth in private and public hospitals in NSW (2000–2008).

**Design:** Linked data population-based retrospective cohort study involving five data sets.

**Setting:** New South Wales, Australia.

**Participants:** 691 738 women giving birth to a singleton baby during the period 2000–2008.

**Main outcome measures:** Rates of neonatal resuscitation, perinatal mortality, neonatal admission following birth and readmission to hospital in the first 28 days of life in public and private obstetric units.

**Results:** Rates of obstetric intervention among low-risk women were higher in private hospitals, with primiparous women 20% less likely to have a normal vaginal birth compared to the public sector. Neonates born in private hospitals were more likely to be less than 40 weeks; more likely to have some form of resuscitation; less likely to have an Apgar <7 at 5 min. Neonates born in private hospitals to low-risk mothers were more likely to have a morbidity attached to the birth admission and to be readmitted to hospital in the first 28 days for birth trauma (5% vs 3.6%); hypoxia (1.7% vs 1.2%); jaundice (4.8% vs 3%); feeding difficulties (4% vs 2.4%) ; sleep/behavioural issues (0.2% vs 0.1%); respiratory conditions (1.2% vs 0.8%) and circumcision (5.6 vs 0.3%) but they were less likely to be admitted for prophylactic antibiotics (0.2% vs 0.6%) and for socioeconomic circumstances (0.1% vs 0.7%). Rates of perinatal mortality were not statistically different between the two groups.

**Conclusions:** For low-risk women, care in a private hospital, which includes higher rates of intervention, appears to be associated with higher rates of morbidity seen in the neonate and no evidence of a reduction in perinatal mortality.

### Strengths and limitations of this study

- The strength of this study lies in the large sample size of linked birth data admissions associated with these births.
- The use of data from five population-based datasets which have been linked to enhance validity and ascertainment.
- Limitations are the restricted number of variables that are included and the scarcity of specific information on potential confounders.
- Body mass index and key sociodemographic risk factors could not be controlled for and this would have added risk to women giving birth in public hospitals.

## INTRODUCTION

In Australia, the national statistics reveal that 29% (n=83 573) of women who gave birth in hospital gave birth in private hospitals directly under private obstetric care.[1] The remaining 71% (n=204 399) of women gave birth in public hospitals in Australia. Women who are privately insured have been reported to have better maternal and perinatal outcomes compared to women who give birth in public hospitals as public patients[2]; but it has been argued that these women tend to be less socioeconomically disadvantaged and healthier[3] and therefore might be expected to have better outcomes. Arguments about the impact of private status on health outcomes are in reality complex.

What is not disputed are the much higher rates of obstetric intervention that occur in private hospitals in Australia. At a national level, the intervention rates in childbirth,

such as caesarean section, are significantly higher in the private sector (43.1% vs 28.4%) and the rates of normal vaginal birth significantly lower (42.7% vs 61%).[1] Despite the rising intervention rates over the past decade, the perinatal mortality rate overall has not shown a corresponding decline.[1] There is also growing concern that the short-term and long-term morbidity associated with major obstetric interventions, such as caesarean, may not be insignificant for the mother[4] and the baby.[5 6] The cost to the tax payer of the rising intervention in childbirth is also significant.[7 8]

A recent study in New South Wales, Australia[9] found among 293 840 low-risk women, rates of obstetric intervention were highest in private hospitals and lowest in public hospitals. Low-risk primiparous women giving birth in a private hospital compared to a public hospital had higher rates of induction (31% vs 23%); instrumental birth (29% vs 18%); caesarean section (27% vs 18%), epidural (53% vs 32%), episiotomy (28% vs 12%) and lower normal vaginal birth rates (44% vs 64%). Low-risk multiparous women had higher rates of instrumental birth (7% vs 3%), caesarean section (27% vs 16%), epidural (35% vs 12%), episiotomy (8% vs 2%) and lower normal vaginal birth rates (66% vs 81%). Following a comparison with data from the decade previously,[10] these interventions were found to have increased by 5% for women in public hospitals and by over 10% for women in private hospitals.[9] Among low-risk primiparous women giving birth in private hospitals, 15/100 women had a vaginal birth with no obstetric intervention compared to 35/100 women giving birth in a public hospital.[9] Concern was expressed that perinatal mortality and morbidity was not reported in that paper.[11] In this study we aim to address this through examining the rates of obstetric intervention and associated perinatal mortality and morbidity attached to the birth admission and readmission to hospital in the first 28 days of life for low-risk women giving birth in private and public hospitals in NSW (2000–2008).

## METHODS
### Data sources
The New South Wales Centre for Health Record Linkage conducted linkage of several datasets via the Health Record Linkage (CHeReL). Pregnancy and birth data for the time period 1 July 2000 until 2 June 2008 of all singleton births were provided by New South Wales (NSW), Ministry of Health as recorded in the NSW midwives data collection (MDC), now the perinatal data collection (PDC). This population-based surveillance system contains maternal and infant data on all births of greater than 400 g birth weight and/or 20 completed weeks gestation. Hospitals are coded either as private or public in the data set. However, the data identifying women who received care in public hospitals under private accommodation status is no longer collected as it had been in the years 1996–1997 and for this reason

patients who are under private obstetric care in public hospitals are not able to be differentiated from their public counterparts, so for this study we analysed the data by hospital (private/public). A previous study published in 2000[10] showed that there was a moderating factor on intervention rates when women with private insurance status gave birth in a public hospital, leading to lower intervention rates than when they gave birth in private hospitals.

The NSW PDC contains statistics on all births in New South Wales—which amounts to one-third of all births which occur in Australia annually. Data is provided for maternal age, maternal hypertension, maternal diabetes, parity, private/public patient status, fetal presentation, onset of labour, gestation at birth, delivery type, Apgar scores and admission to neonatal intensive care and resuscitation details for the neonate. This dataset (NSW PDC) was linked to the Admitted Patient Data Collection (APDC) for the time period 1 July 2000 to 30 June 2008. The APDC records all admitted patient services provided by NSW Public Hospitals, Public Psychiatric Hospitals, Public Multi-Purpose Services, Private Hospitals and Private Day Procedures Centres. The APDC provided additional information, such as data on maternal medical conditions, which was used to exclude further maternal cases and was used to calculate admission and readmission details for neonates. Further linkage occurred in the NSW Registry of Births, Deaths and Marriages (RBDM) and the Australian Bureau of Statistics Death Data, which provided mortality data. The NSW Register of congenital conditions provided cases of congenital conditions, as did the coding in the APDC. Any neonate (and mother pair) with a recorded congenital condition (ICD-10-AM codes Q0.0-Q99.9) on either dataset was removed from the dataset due to their high-risk status. Probabilistic data linkage techniques were utilised for data linkage and de-identified datasets were provided for analysis. Probabilistic record linkage software assigns a 'linkage weight' to pairs of records. For example, records that match perfectly or nearly perfectly on first name, surname, date of birth and address have a high linkage weight and records that match only on date of birth have a low-linkage weight. If the linkage weight is high it is likely that the records truly match, and if the linkage weight is low it is likely that the records are not truly a match. This technique has been shown to have a false-positive rate of 0.3% of records.[12]

Gestation is recorded at birth and is also recorded in the database according to the woman's menstrual history, usually combined with a routine scan at 12–13 weeks.

Admission to neonatal intensive care refers to admission to special care nursery (SCN) or neonatal intensive care unit (NICU).

Any resuscitation includes suction of the mouth or nostrils at birth; oxygen administered by mask; intermittent positive pressure respiration (IPPR) by bag and mask or by intubation; external cardiac massage and ventilation.

## Subjects

We classified the low-risk primipara as a first time mother aged 20–34 years, who had no pre-existing or pregnancy-related hypertension or diabetes, was a non-smoker and gave birth at 37–41 completed weeks gestation to a singleton baby in a cephalic presentation within the 10th and 90th centiles for gestation and birth weight. The low-risk multipara was a woman aged 20–34 years having her second or subsequent baby, who had no pre-existing or pregnancy-related hypertension or diabetes, was a non-smoker, gave birth at 37–41 completed weeks gestation to a singleton baby in a cephalic presentation within the 10th and 90th centiles for gestation and birth weight. We excluded women with a previous caesarean section or who were induced for a medical indication, or who underwent a caesarean section for a pre-existing medical indication or gave birth without a trained birth attendant (born before arrival). If a caesarean section was undertaken during labour however for non-reassuring heart rate, dystocia, etc these women were included in the study. These characteristics were defined first from the PDC with additional medical conditions identified in the APDC being used to exclude cases.

## Outcomes

Any neonatal admission including the ICD-10-AM codes Z37.0 (single live birth), Z37.1 (single stillbirth) or Z38.0 (singleton born in hospital) was deemed the birth admission and any ICD-10-AM codes referring to conditions which arise in the perinatal period (P00-P96) and those referring to factors influencing health status and contact with health services (Z00-Z99) which were included in this admission were deemed morbidities associated with the birth admission. Any other admission following this discharge from the initial birth admission to home or another hospital was deemed a readmission and included transfers to a hospital other than that where the birth occurred. When examining readmission data, all ICD-10-AM codes recorded were reviewed and those where ≥10 events occurred in either private or public hospitals were marked for analysis. Events were grouped in body systems where appropriate or under headings such as infection for ease of analysis and interpretation.

Morbidity was recorded with the birth admission and rates of events were calculated using the number of babies who had any morbidity recorded with their birth.

Stillbirth and neonatal deaths were calculated from multiple sources but were limited to those that occurred within 28 days of birth and they were only counted once. Death may have been detected on any one of the following four datasets. The PDC 'Discharge status' variable or admissions in the APDC where the case mode separation was coded as 'Died' or the NSW RBDM or ABS Death Data where a death had been recorded. The maternal admission data for any admission that occurred during the pregnancy, as well as the birth admission for all cases of stillbirth or neonatal death were examined to determine any maternal medical or pregnancy-related condition. This methodology of utilising multiple data sources to identify cases has been shown by Lain et al[13] to be the most reliable way to increase ascertainment of cases.

Obstetric intervention was defined to include induction, epidural use, episiotomy, instrumental delivery (requiring the use of forceps or vacuum) and delivery via caesarean section.

## Data analysis

The cohort was divided into primiparous and multiparous women for the primary analysis of birth outcomes. When examining neonatal status at birth mortality ORs were calculated using logistic regression with and without adjustment for age and gestation. For neonatal morbidity at birth and readmission, $\chi^2$ statistics were calculated for observed events. The total number of babies born in a public or a private hospital were used as the denominator when calculating the percentage of babies born with a morbidity code attached to their birth record or the number of babies readmitted with a designated morbidity code. This methodology provides for comparison between place of birth taking into consideration the fact that up to 55 morbidity codes can be attached to any one birth or readmission record. Taking into account the size of the cohort and the number of analyses undertaken, results were considered significant at the level $p<0.01$. Analysis was undertaken with IBM SPSS V.20.

## RESULTS
### Maternal characteristics, interventions and outcomes

The PDC dataset for the time period 1 July 2000 to 2 June 2008 contained the antenatal, birth and postnatal details on 691 738 births. The APDC for the time period 1 July 2000 to 30 June 2008 contained >1.1 million admissions for the neonates/children of these women.

From the total population of primiparous women (288 309 women), 29 597 low-risk primiparous women gave birth in private hospitals in NSW and 79 792 low-risk primiparous women gave birth in public hospitals. The rates of obstetric intervention were much higher among those who gave birth in private hospitals compared to those who gave birth in public hospitals when all interventions for prespecified medical reasons were removed. Low-risk primiparous women giving birth in private hospitals compared to low-risk primiparous women giving birth in public hospitals had higher rates of induction for non-medical reasons (19% vs 7%), instrumental birth (30% vs 20%), caesarean section (25% vs 16%), epidural (71% vs 35%) and episiotomy (42% vs 23%). Severe perineal trauma (defined as third-degree and fourth-degree perineal trauma) was lower in a private hospital in first-time mothers (4.7% vs 5.4%; table 1).

Among the total population of multiparous women (403 429 women), 28 703 low-risk multiparous women gave birth in private hospitals and 99 212 low-risk

| Table 1 Maternal characteristics, interventions and outcomes for low-risk primiparous women in New South Wales (2000–2008) | | |
|---|---|---|
| **Low-risk primiparous women** | **Private hospital (n=29 597) (%)** | **Public hospital (n=79 792) (%)** |
| Maternal age (years) | | |
| 20–24 | 6.2 | 28.9 |
| 25–29 | 39.9 | 40.9 |
| 30–34 | 53.9 | 30.2 |
| Weeks gestation at delivery | | |
| 37 | 4.5 | 4.3 |
| 38 | 15.4 | 11.8 |
| 39 | 27.8 | 25.0 |
| 40 | 43.2 | 39.7 |
| 41 | 9.1 | 19.2 |
| Type of labour | | |
| Spontaneous | 71.9 | 89.9 |
| Induced | 19.2 | 7.1 |
| No labour | 8.9 | 3.0 |
| Delivery | | |
| Normal vaginal | 44.9 | 64.8 |
| Forceps | 11.5 | 6.7 |
| Vacuum | 18.9 | 12.9 |
| Total caesarean section | 24.7 | 15.6 |
| Caesarean section (after labour) | 15.9 | 12.6 |
| Caesarean section before the onset of labour | 8.8 | 3.0 |
| Epidural | 70.8 | 35.4 |
| Episiotomy | 42.4 | 23.3 |
| Severe perineal trauma | 4.7 | 5.4 |

| Table 2 Maternal characteristics, interventions and outcomes for low-risk multiparous women in New South Wales (2000–2008) | | |
|---|---|---|
| **Low-risk multiparous women** | **Private hospital (n=28 703) (%)** | **Public hospital (n=99 212) (%)** |
| Maternal age (years) | | |
| 20–24 | 2.1 | 16.8 |
| 25–29 | 25.8 | 38.8 |
| 30–34 | 72.1 | 44.4 |
| Weeks gestation at delivery | | |
| 37 | 4.1 | 4.0 |
| 38 | 18.7 | 13.0 |
| 39 | 31.6 | 26.9 |
| 40 | 40.2 | 40.8 |
| 41 | 5.4 | 15.3 |
| Type of labour | | |
| Spontaneous | 64.0 | 87.4 |
| Induced | 32.1 | 10.1 |
| No labour | 3.9 | 2.5 |
| Delivery | | |
| Normal vaginal | 86.1 | 92.7 |
| Forceps | 1.9 | 0.7 |
| Vacuum | 6.1 | 2.1 |
| Total caesarean section | 5.9 | 4.5 |
| Caesarean section after labour | 2.0 | 2.0 |
| Caesarean section before the onset of labour | 3.9 | 2.5 |
| Epidural | 34.4 | 9.5 |
| Episiotomy | 16.2 | 5.1 |
| Severe perineal trauma | 0.9 | 0.9 |

multiparous women gave birth in public hospitals. The rate of obstetric intervention was significantly higher among those who gave birth in private hospitals in NSW compared to those who gave birth in public hospitals when all interventions for specific medical reasons were removed. Low-risk multiparous women who gave birth in private hospitals compared to low-risk multiparous women giving birth in public hospitals had higher rates of induction for non-medical reasons (32% vs 10%), instrumental birth (8% vs 3%), epidural (34% vs 10%) and episiotomy (16% vs 5%) and similar rates of severe perineal trauma (0.9%). The caesarean section rate still remained higher in the private cohort (5.9% vs 4.5%) though this was mostly associated with elective caesarean section (table 2).

### Perinatal characteristics, interventions and outcomes
There was no difference in birth weight between babies born in a private and public hospital. Babies born in a private hospital were more likely to be born at 37, 38, 39 and 40 weeks and less likely to be born at 41 weeks gestation (figure 1).

Babies of primiparous women who gave birth in a private hospital were less likely to have an Apgar of <7 at 5 min (adjusted OR (aOR) 1.34 95% CI 1.18 to 1.53; p<0.001) as were babies of multiparous women who gave birth in private hospitals (aOR 1.37 95% CI 1.14 to 1.64;

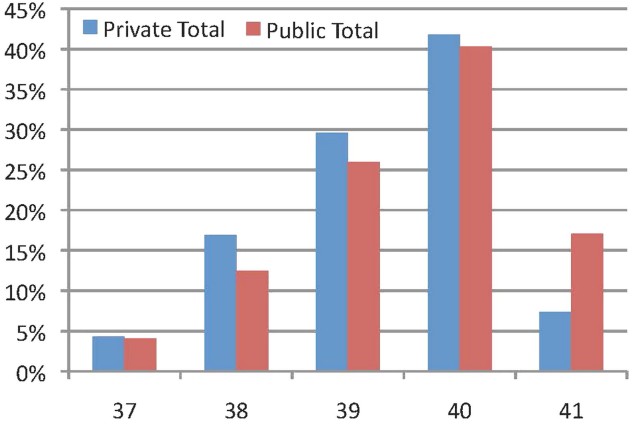

**Figure 1** Comparison of gestation at delivery between all low-risk women stratified by hospital type.

**Table 3** Perinatal outcomes adjusted for maternal age and gestation at birth for low-risk primiparous women

| | Private (n=29 597) | Public (n=79 791) | OR* | aOR* | p Value |
|---|---|---|---|---|---|
| Apgar <7 at 5 min | 296 (1.0%) | 1037 (1.3%) | 1.36 (1.12–1.54) | 1.34 (1.18–1.53) | <0.001 |
| Any resuscitation† | 18 498 (62.5%) | 30 560 (38.3%) | 0.372 (0.36–0.38) | 0.364 (0.36–0.37) | <0.001 |
| Admitted to SCN and/or NICU | 3078 (10.4%) | 8139 (10.2%) | 1.00 (0.96–1.05) | 1.03 (0.98–1.08) | 0.210 |
| Transferred | 178 (0.6%) | 3351 (4.2%) | 7.30 (6.29–8.40) | 7.55 (6.52–8.74) | <0.001 |
| Total perinatal mortality | 22 (0.74/1000) | 85 (1.06/1000) | 1.40 (0.93–2.01) | 1.49 (0.93–2.41) | 0.100 |

*Private hospital is the reference category.
†Any resuscitation includes: suction, oxygen, intermittent positive pressure respiration by bag and mask, intubation and IPPR, external cardiac massage and ventilation and other.
NICU, neonatal intensive care unit; SCN, special care nursery.

p<0.001). Babies born in private hospitals were less likely to have no resuscitation (aOR 0.36 95% CI 0.35 to 0.37; p<0.001). Babies born to low-risk primiparous women in a private hospital were no more likely to be admitted to special care and/or neonatal intensive care (aOR 1.03 95% CI 0.98 to 1.08; p 0.210) and were less likely to have their baby transferred to another hospital (aOR 7.55 95% CI 6.52 to 8.74; p<0.001). There was no difference in the perinatal mortality rate for babies of primiparous women born in private or public hospitals (aOR 1.49 95% CI 0.93 to 2.41; p=0.10, table 3). Similar outcomes were seen for babies born to multiparous women in private and public hospitals (table 4).

### Reason for birth admission of neonates
We examined neonatal morbidity as coded on the neonatal birth admission record and found fewer babies overall had a morbidity recorded (ICD-10-AM code other than the birth code) in the private sector compared to the public sector (40.0% vs 53.9%) due to the added clinical and social maternal complexity in the public sector. There were however some significant differences noted under the main ICD-10-AM Grouping *Certain conditions originating in the perinatal period (P00-P96)*. Babies born in a private hospital were more likely to have been affected by a forceps or vacuum delivery and were more likely to have trauma to the scalp (3.22% vs 2.22%), intrauterine hypoxia (1.70% vs 1.21%), jaundice (4.68% vs 2.89%), minor cardiac murmurs (0.24% vs 0.17%), conjunctivitis (2.98% vs

1.27%), respiratory conditions (0.96% vs 0.57%), temperature regulation issues (2.08% vs 1.27%), feeding difficulties (3.83% vs 2.32%), carbohydrate metabolism issues (0.86% vs 0.53%), vomiting (0.55% vs 0.39%) and to be circumcised during the birth admission if a male (5.31% vs 0.21%). Babies born in a public hospital were more likely to be admitted for observation and evaluation (6.35% vs 3.75%) have prophylactic antibiotics (0.52% vs 0.16%) and be admitted for socioeconomic circumstances (eg, housing, distance, adoption and assumption of care) (0.57% vs 0.05%) (table 5).

### Reason for birth readmission of neonate up to 28 days of age
We examined the reasons for transfer or readmission of babies up until 28 days of age and found, that though the numbers are small, more babies born in private hospitals were readmitted compared to babies born in a public hospital (0.95% vs 0.65%; table 6). Babies born in private hospitals were more likely to be readmitted for infectious diseases (0.21% vs 0.12%) endocrine, nutritional and metabolic disorders (0.05% vs 0.02%), sleep disorders (0.03% vs 0.01%), hypoglycaemia (0.02% vs 0.01%), birth trauma such as cephalohaematoma (0.02% vs 0.01%), trauma involving the scalp (0.05% vs 0.02%), excessive crying (0.14% vs 0.07%), behavioural disorders (0.06% vs 0.02%) and for circumcision if a male (0.20 vs 0.13). Babies born in public hospitals were more likely to be readmitted with respiratory disorders (0.27% vs 0.20%), injury and poisoning (eg,

**Table 4** Perinatal outcomes adjusted for maternal age and gestation at birth for low-risk multiparous women

| | Private (n=28 703) | Public (n=99 212) | OR* | aOR* | p Value |
|---|---|---|---|---|---|
| Apgar <7 at 5 min | 149 (0.5%) | 676 (0.7%) | 1.32 (1.10–1.57) | 1.37 (1.14–1.64) | <0.001 |
| Any resuscitation† | 14 820 (51.6%) | 29 867 (30.1%) | 0.404 (0.39–0.42) | 0.399 (0.39–0.46) | <0.001 |
| Admitted to SCN and/or NICU | 1775 (6.2%) | 5870 (5.9%) | 0.957 (0.91–1.01) | 1.027 (0.97–1.09) | 0.363 |
| Transferred | 232 (0.8%) | 4375 (4.4%) | 5.661 (4.96–6.47) | 6.516 (5.70–7.45) | <0.001 |
| Total perinatal mortality | 17 (0.59/1000) | 76 (0.77/1000) | 1.294 (0.77–2.19) | 1.294 (0.75–2.23) | 0.355 |

*Private hospital is the reference category.
†Any resuscitation includes: Suction, oxygen, intermittent positive pressure respiration by bag and mask, Intubation and IPPR, external cardiac massage and ventilation and other.
NICU, neonatal intensive care unit; SCN, special care nursery.

**Table 5** Morbidity associated with birth admission coded on neonatal birth admission record

| ICD-10-AM Grouping | Private n=58 300 | | Public n=179 003 | | p Value |
|---|---|---|---|---|---|
| | Count | Per cent | Count | per cent | |
| **Certain conditions originating in the perinatal period (P00–P96)** | | | | | |
| Fetus and newborn affected by maternal infectious and parasitic diseases (P00.2) | 41 | 0.07 | 989 | 0.55 | <0.001 |
| Fetus and newborn affected by forceps delivery (P03.2) | 473 | 0.81 | 1108 | 0.62 | <0.001 |
| Fetus and newborn affected by delivery by vacuum extractor (P03.3) | 511 | 0.88 | 1509 | 0.84 | 0.46 |
| **Birth trauma (all body systems) (P10–P15)** | **2948** | 5.06 | **6447** | 3.60 | <0.001 |
| specifically to scalp (included in above total) (P12) | 1880 | 3.22 | 3965 | 2.22 | <0.001 |
| Intrauterine hypoxia (P20) | 993 | 1.70 | 2170 | 1.21 | <0.001 |
| Other specified respiratory conditions of newborn (P28) | 562 | 0.96 | 1015 | 0.57 | <0.001 |
| Benign and innocent cardiac murmurs in newborn (P29.82) | 139 | 0.24 | 303 | 0.17 | 0.001 |
| Neonatal conjunctivitis specific to the perinatal period (P39.1) | 1740 | 2.98 | 2267 | 1.27 | 0.001 |
| **Jaundice-related conditions (P58–P59)** | **2728** | 4.68 | **5166** | 2.89 | <0.001 |
| Transitory disorders of carbohydrate metabolism specific to fetus and newborn (P70) | 502 | 0.86 | 942 | 0.53 | <0.001 |
| Conditions involving the integument and temperature regulation of fetus and newborn (P80–P83) | 1214 | 2.08 | 2275 | 1.27 | <0.001 |
| Vomiting in newborn (P92.0) | 320 | 0.55 | 693 | 0.39 | <0.001 |
| Feeding problems in newborn (P92) | 2231 | 3.83 | 4157 | 2.32 | <0.001 |
| **Factors influencing health status and contact with health services (Z00–Z99)** | | | | | |
| Routine and ritual circumcision (Z41.2)* | 1552 | 5.31 | 187 | 0.21 | <0.001 |
| Observation and evaluation of newborn (Z03) | 2187 | 3.75 | 11 372 | 6.35 | <0.001 |
| Prophylactic chemotherapy (antibiotics) (Z29.2) | 93 | 0.16 | 935 | 0.52 | <0.001 |
| Socioeconomic circumstances (housing, distance, adoption, assumption of care) (Z76) | 32 | 0.05 | 1020 | 0.57 | <0.001 |

Bold typeface denotes where several similar codes have been combined; non-bold indicates results are for one code.
*As a percentage of male babies.

burns) (0.05% vs 0.03%), antibiotic therapy (0.03% vs 0.01%) and socioeconomic circumstances (housing, distance, adoption, assumption of care (0.09% vs 0.04%).

### Combined birth and readmission neonatal morbidity for selected codes

When we combined major birth and readmission morbidities for key selected codes we found that in the first 28 days following birth, babies born in private hospitals were significantly more likely to be admitted for feeding difficulties (4% vs 2.4%), circumcision if a male (5.6 vs 0.3), birth trauma (mostly scalp trauma) (5% vs 3.6%), jaundice (4.8% vs 3.0%), hypoxia (1.7% vs 1.2%), respiratory disorders (1.2% vs 0.8%) and sleep/behavioural issues (0.2% vs 0.1%). Babies born in public hospitals were more likely to be admitted for socioeconomic circumstances such as housing, distance, adoption or assumption of care (0.7% vs 0.1%) and prophylactic antibiotics (0.6% vs 0.2%) (table 7 and figure 2).

### DISCUSSION
### Intervention rates

Despite being an extremely low-risk cohort, less than half the primiparous women in this study giving birth in a private hospital had a normal vaginal birth (45% vs 65%); this was 20% lower than in the public cohort. One in five primiparous women giving birth in a private hospital were induced and nearly one in two had an episiotomy. For low-risk multiparous women giving birth in a private hospital nearly one in three were induced. The trend for higher intervention rates has been reported for low-risk women giving birth in the private sector in Australia previously and continues to show an increase.[9 14 15] In a recent publication we showed that the rate of caesarean section had increased in both the private and public sector in the past decade in low-risk women.[9] It has been argued in a previous publication that these high intervention rates in the private sector led to better perinatal outcomes than in the public sector.[2] This publication received significant criticism in letters to the editor[3 16 17] for several methodological flaws, including most significantly the failure to adjust for low-birth weight, inadequate ascertainment of congenital abnormalities and failure to look at perinatal morbidity. In this study we included only low-risk women, adjusting for maternal age and gestational age differences. We also removed all babies with congenital abnormalities from this data set. We found that the perinatal mortality rate was not statistically different when the populations were matched in this data set for maternal risk.

**Table 6** Morbidity associated with readmission of the baby ≤28 days of age

| ICD-10-AM Grouping | Private n=58 300 | | Public n=179 003 | | |
|---|---|---|---|---|---|
| | Count | Per cent | Count | Per cent | p Value |
| Certain infectious and parasitic diseases (A00–B99) | **121** | 0.21 | **217** | 0.12 | <0.001 |
| Endocrine nutritional and metabolic diseases (E00–E89) | **30** | 0.05 | **38** | 0.02 | <0.001 |
| Volume depletion (E86) | 18 | 0.03 | 12 | 0.01 | <0.001 |
| Mental and behavioural disorders (F00–F99) | **33** | 0.06 | **30** | 0.02 | <0.001 |
| Non-organic hypersomnia (F51.1) | 17 | 0.03 | 19 | 0.01 | 0.002 |
| Diseases of the nervous system (G00–G99) | **34** | 0.06 | **38** | 0.02 | <0.001 |
| Disorders of the sleep wake schedule (G47.2) | 18 | 0.03 | 9 | 0.01 | <0.001 |
| Diseases of the ear and mastoid process (H60–H95) | **19** | 0.03 | **23** | 0.01 | 0.002 |
| Diseases of the respiratory system (J00–J99) | **155** | 0.27 | **351** | 0.20 | 0.002 |
| Acute obstructive laryngitis (croup) (J05) | 6 | 0.01 | 16 | 0.01 | 0.96 |
| Acute upper respiratory infection unspecified (J06) | 16 | 0.03 | 52 | 0.03 | 0.96 |
| Pneumonia (J10–J18) | 6 | 0.01 | 19 | 0.01 | 0.96 |
| Acute bronchiolitis (J21) | 70 | 0.12 | 175 | 0.10 | 0.17 |
| Unspecified acute lower respiratory tract infection (J22) | 3 | 0.01 | 12 | 0.01 | * |
| Diseases of the digestive system (K00–K93) | **53** | 0.09 | **92** | 0.05 | 0.001 |
| Gastro-oesophageal reflux disease (K21) | 26 | 0.04 | 37 | 0.02 | 0.003 |
| Certain conditions originating in the perinatal period (P00-P96) | **474** | 0.81 | **1011** | 0.56 | <0.001 |
| Cephalohaematoma due to birth trauma (P12.0) | 14 | 0.02 | 13 | 0.01 | 0.002 |
| Total birth trauma to scalp (P12) | 30 | 0.05 | 32 | 0.02 | <0.001 |
| Intrauterine hypoxia (P20) | 4 | 0.01 | 25 | 0.01 | * |
| Other neonatal hypoglycaemia (P70.4) | 14 | 0.02 | 14 | 0.01 | 0.002 |
| Other transitory neonatal electrolyte and metabolic disturbances (P70.8) | 5 | 0.01 | 21 | 0.01 | 0.527 |
| Fever of newborn (P81.9) | 5 | 0.01 | 18 | 0.01 | 0.753 |
| Feeding problems of newborn (P92) | 40 | 0.07 | 100 | 0.06 | 0.271 |
| **Neonatal jaundice (P58)** | **193** | 0.33 | **90** | 0.05 | <0.001 |
| Symptoms, signs and abnormal findings not elsewhere classified (R00–R99) | **198** | 0.34 | **340** | 0.19 | <0.001 |
| Fever (R50) | 19 | 0.03 | 27 | 0.02 | 0.008 |
| Feeding difficulties and mismanagement (R63.3) | 25 | 0.04 | 37 | 0.02 | 0.003 |
| Excessive crying (R68.1) | 83 | 0.14 | 117 | 0.07 | <0.001 |
| Injury, poisoning and certain other consequences of external causes (S00–T98) | **15** | 0.03 | **94** | 0.05 | 0.009 |
| Burns (T20–T31) | 0 | 0.00 | 30 | 0.02 | * |
| Factors influencing health status and contact with health services (Z00–Z99) | **192** | 0.33 | **691** | 0.39 | 0.051 |
| Observation and evaluation in newborn (Z03) | 34 | 0.06 | 102 | 0.06 | 0.907 |
| Prophylactic chemotherapy (antibiotics) (Z29.2) | 5 | 0.01 | 49 | 0.03 | 0.009 |
| Routine and ritual circumcision (Z41.2) | 68 | 0.20 | 119 | 0.13 | <0.001 |
| Attention to surgical dressings and sutures (Z48.0) | 0 | 0.00 | 28 | 0.02 | * |
| Socioeconomic circumstances (housing, distance, adoption, assumption of care) (Z76) | 22 | 0.04 | 156 | 0.09 | <0.001 |

Bold typeface denotes where several similar codes have been combined; non-bold indicates results are for one code.
*Cell size too small to calculate $\chi^2$.
†As a percentage of male babies.

## Neonatal resuscitation and admission to SCN/NICU

We found that babies born in a private hospital were much more likely to experience some form of resuscitation, in particular twice the rate of suctioning at birth. Routine suctioning for infants born with clear and/or meconium stained amniotic fluid is not recommended[18] as it can cause a bradycardia[19] and there is no evidence of benefit. We are unsure why such a high rate of newborn suctioning continues in the private sector. Rates of Apgar scores of ≤7 at 5 min were slightly higher among low-risk women who gave birth in public hospitals, and this has been demonstrated in another recent Australian publication,[15] overall the babies were no more likely to be admitted to SCN/NICU compared to babies born in private hospitals.

## Neonatal admission and readmission

We found some interesting differences in morbidity however when examining morbidity attached to the birth admission and readmission to hospital in the first

**Table 7** Combined birth and readmission neonatal morbidity for selected codes

|  | Private | Public | p Value* |
|---|---|---|---|
| Total feeding difficulties | 2314 (4.0%) | 4306 (2.4%) | <0.0001 |
| Total circumcision† | 1620 (5.6%) | 306 (0.3%) | <0.0001 |
| Total socioeconomic circumstances | 54 (0.1%) | 1176 (0.7%) | <0.0001 |
| Total birth trauma | 2922 (5.0%) | 6492 (3.6%) | <0.0001 |
| Total hypoxia | 997 (1.7%) | 2195 (1.2%) | <0.0001 |
| Total jaundice | 2818 (4.8%) | 5359 (3.0%) | <0.0001 |
| Total respiratory | 717 (1.2%) | 1366 (0.8%) | <0.0001 |
| Total sleep/ behavioural issues | 118 (0.2%) | 145 (0.1%) | <0.0001 |
| Prophylactic antibiotics | 98 (0.2%) | 982 (0.6%) | <0.001 |

*$\chi^2$ square.
†As a percentage of male babies.

28 days for codes that may be associated with the higher rates of obstetric intervention in the private sector and a different sociodemographic profile in the public sector. While increasingly preterm babies >35 weeks/ and or >2.2k and some cases of jaundice may be managed at the bedside in some hospitals, this is less likely to occur in a private hospital. It is more likely to occur in large maternity units.

Birth trauma, in particular injuries to the scalp, were significantly more common in the private sector and these are generally associated with instrumental birth, including vacuum extraction.[20–22] With more women (nearly one in three primiparous women) experiencing

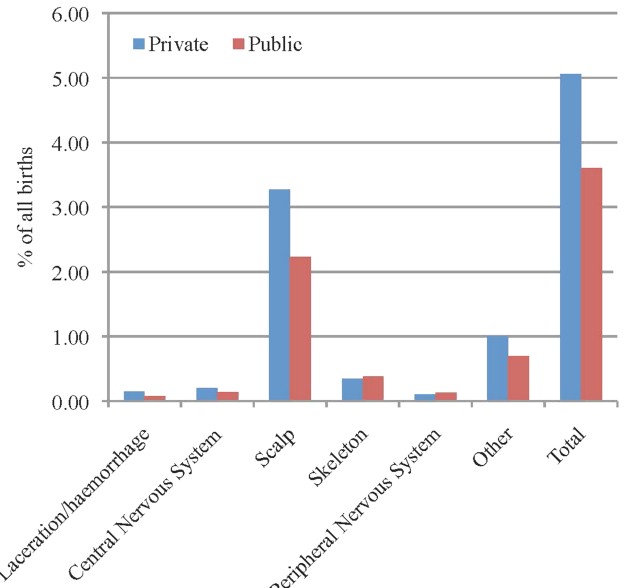

**Figure 2** Birth trauma as a percentage of all births in private and public hospitals.

an instrumental birth in the private cohort and one in five women in the public sector this is not surprising. Birth trauma is associated with a longer hospital stay and increased risk of admission to SCN/NICU as well as higher rates of neonatal morbidity including neurological morbidity (hypotonia, jitteriness, convulsions and hypoxic ischaemic encephalopathy) and jaundice.[22]

Jaundice was observed to be higher in the private sector, which may be related to several factors, such as the elective delivery of babies at an earlier gestation, the increased scalp trauma due to the high instrumental birth rate, as discussed above and potential breastfeeding difficulties due to higher use of epidural analgesia. Jaundice has been associated with birth trauma, in particular delivery by vacuum extraction and problems with feeding, especially supplementary feeding.[22 23] Earlier gestational age <39 weeks has also been found to be associated with jaundice, with this decreasing with each week of additional gestation.[24] The gestational age of babies born in private hospitals in this study was significantly lower than in the public sector possibly due to the high rates of non-medical induction of labour and non-medically indicated caesarean section before the onset of labour.

While there have been studies associating difficulties with breastfeeding and higher rates of jaundice, the recent publication from the Universal Screening for Hyperbilirubinemia Study Group found difficulties with breastfeeding was a minimal risk factor.[24]

Nearly twice as many babies who were born in a private hospital in this study were admitted or readmitted with feeding problems compared to babies born in a public hospital. Feeding difficulties are associated with operative birth interventions and being early term.[25 26] Breastfeeding outcomes are positively associated with uncomplicated unassisted vaginal birth where the mother and infant remain together and breastfeeding is started within an hour of the birth and following skin-to-skin contact. Interventions during labour and birth can impact on the initiation and duration of breastfeeding. Caesarean section,[27–30] instrumental birth,[31] epidural anaesthesia and opioid analgesia[32–34] use have all been associated with breastfeeding difficulties. All these birth interventions were higher in the private cohort in this study. In addition early term birth, which is mainly due to induction of labour and elective caesarean section[35 36] is associated with increased breastfeeding difficulties along with other serious morbidities.[37] Unmedicated newborns are more highly aroused immediately following the birth[38] and able to breastfeed without assistance if given skin to skin contact and freedom from intrusive procedures.[39 40] Following caesarean section there can be a significantly longer period of time until a mother touches and holds her newborn compared to an unassisted vaginal birth.[41]

In a previous paper[6] using national Australian population data we found that among low-risk women who had an unassisted vaginal birth with spontaneous onset of labour and no labour augmentation, the odds of

admission to neonatal intensive care or special care nursery were significantly increased when the baby was 37 weeks gestation at the time of birth compared to later gestations. Some claim that during the final weeks of gestation the fetal brain goes through a marked increase in mass and nerve growth (corticoneurogenesis) which may be best left undisturbed by allowing the normal gestational length to occur.[42] In this study low-risk women giving birth in private hospitals in NSW were much more likely to give birth at earlier gestations than their public hospital counterparts for every week up to and including 40 weeks, but they were significantly less likely to deliver at 41 weeks. This may also help to explain why more babies born in a private hospital were readmitted with, respiratory, feeding, jaundice and sleep and behavioural problems. However, there is also evidence that there are increased adverse perinatal outcomes for babies born following 41 completed weeks, but we did not examine this population.[43]

## Circumcision

Babies born in a private hospital were significantly more likely to be circumcised in the first 28 days of life. This may be due to different information being given in private hospitals about the procedure or easy access to providers who perform the procedure. Circumcision rates are estimated to be between 10% and 20% in Australia[44] and are decreasing. A recent position statement of the Royal Australian College of Physicians states "that the frequency of diseases modifiable by circumcision, the level of protection offered by circumcision and the complication rates of circumcision do not warrant routine infant circumcision in Australia and New Zealand."[44]

## Socioeconomic circumstances

The difference in the socioeconomic status of the women giving birth in public compared to private hospitals appears to be demonstrated by the significantly higher rates of public hospital babies with a morbidity attached to the birth admission or readmission in the first 28 days for socioeconomic circumstances, including housing, distance, adoption and assumption of care. This again confirms what is already known that the two populations are very different sociodemographically with a greater disadvantage in the public sector.

## Limitations

Our study is limited to providing a snapshot of perinatal outcomes in the most populous state in Australia in a defined time period for women who have no indicated risk at birth. However, this study provides useful data following on from our previous paper looking at obstetric intervention in private and public hospitals in NSW providing the reader with a detailed picture of perinatal mortality and morbidity. The advantages of using population-based datasets such as the PDC and the linkage to four other population-based databases include the size of the sample and the high level of accuracy of a validated dataset. The limitations are the restricted number of variables that are included and the scarcity of specific information on potential influencing variables. A small number of cases with a low linkage (false/positive) rate (0.3%) were not included and so there is the possibility of missing adverse outcomes. A previous study showed where stillbirths are excluded due to low linkage these are at lower gestational ages and not term infants as were the focus in this study.[45] Previous validation studies have reported high levels of data accuracy for the majority of diagnoses and procedures conducted during labour and delivery in the state-wide data base,[46 47] although the recording of medical conditions and smoking are overall generally under-reported.[46 48] Having a linked data set provides a much richer picture than we have had previously of the morbidity and mortality associated with birth interventions. While we could not control for obesity due to lack of data, women who have private health insurance have lower rates of obesity and higher socioeconomic status, hence these health disadvantages are most likely over-represented in the women who use public services.[49] There are also several other sociodemographic factors we could not control for, such as education and income, that increase risk for the women giving birth in public hospitals. This study can only provide an overview of possible associations between obstetric interventions and neonatal outcomes and does not imply causality, which could be better obtained from prospective cohort studies.

## CONCLUSION

The continual rise in obstetric intervention for low-risk women in Australia, especially in private hospitals, may be contributing to increased morbidity for healthy women and babies and higher cost of healthcare. The fact that these procedures which were initially life-saving are now so commonplace and do not appear to be associated with improved rates of perinatal mortality or morbidity demands close review. Early term delivery and instrumental births may be associated with increased morbidity in neonates and this requires urgent attention. Previous claims that high-intervention rates in private hospitals lead to better perinatal outcomes than those seen in public hospitals need to be questioned.

**Author affiliations**
[1]Family and Community Health Research Group, School of Nursing and Midwifery, University of Western Sydney, Penrith, New South Wales, Australia
[2]Royal Hospital for Women, University of Sydney, Sydney, Australia
[3]Centre for Newborn Care, Westmead Hospital, Westmead, New South Wales, Australia
[4]School of Medicine, University of Sydney, Camperdown, New South Wales, Australia
[5]Royal Hospital for Women, Randwick, New South Wales, Australia
[6]School of Women and Children's Health, University of NSW, Randwick, New South Wales, Australia
[7]NHMRC Clinical Trials Centre, University of Sydney, Camperdown, New South Wales, Australia

**Contributors** HGD led the study and wrote the paper. ST helped in constructing the study design and writing the paper. MT helped in

constructing the study design. AB helped in writing the paper. CB gave biostatistical support and helped in writing the paper. CT helped with study design, analysis and writing of the paper.

**Funding** This research received no specific grant from any funding agency in the public, commercial or not-for-profit sectors.

**Competing interests** None.

**Ethics approval** Ethical approval was obtained from the NSW Population and Health Services Research Ethics Committee, Protocol No.2010/12/291.

**Provenance and peer review** Not commissioned; externally peer reviewed.

**Data sharing statement** No additional data are available.

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
