## [Reviewer comments · BMJ Open]

Some articles will have been accepted based in part or entirely on reviews undertaken for other BMJ Group journals. These will be reproduced where possible.

ARTICLE DETAILS

TITLE (PROVISIONAL)	Rates of obstetric intervention and associated perinatal mortality and morbidity amongst low risk women giving birth in private and public hospitals in NSW (2000-2008): A linked data population based cohort study.
AUTHORS	Dahlen, Hannah; Tracy, Sally; Tracy, Mark; Bisits, Andrew; Brown, Chris; Thornton, Charlene

VERSION 1 - REVIEW

REVIEWER	Mary-Ann Davey Judith Lumley Centre, La Trobe University, Australia
REVIEW RETURNED	02-Dec-2013

GENERAL COMMENTS	This will be a really valuable contribution to the literature around public versus private maternity care. I have a number of questions/suggestions. Once these are addressed the responses to the questions above are all likely to be 'Yes'. Title: Please change 'cohort' to 'retrospective cohort' Page 6, line 10. Is it known who is and is not 'privately insured'? If they are not analysed by 'insurance' but by 'admission status', they should be referred to as "women admitted as private patients" or similar throughout the paper. In Australia, some women are privately insured, but choose not to use this insurance for maternity care (and are admitted as public patients), and others choose to be admitted as private patients despite not being insured. Most perinatal data collects the admission status, not the insurance status. Page 7, line 5: The paper states that 'these interventions were found to have increased by 5%...'. Perhaps provide a range as they are unlikely to have uniformly increased by 5%. Page 7, line 22: Perhaps there is a word missing, but what is the 'New South Wales Centre'? Page 7, line 34: Were re-admissions recorded in the APDC in July 2008 included to detect those in the first 28 days for births that occurred in June 2008? Page 8, line 16: Please describe how cases below a certain linkage weight were managed in this study. If they were excluded, is there a risk that these would be more or less likely to be cases of mortality or morbidity? Page 8, line 20: Gestation is recorded more than one way - please specify which version was used in this study. Page 8, line 24: It might help an international audience to describe the sorts of conditions that are (and are not) usually managed on the postnatal ward. Page 8, line 38 and Page 9, line 5: add the word 'singleton'. Page 8, lines 7-11: this needs more explanation and justification.
--

e.g. were babies born by caesarean section because of non-reassuring fetal status following a syntocinon infusion excluded? This might artificially reduce the apparent rate of induction, and the rate of CS etc. Which conditions collected from the APDC led to exclusion of cases? Were all women with these conditions excluded?

Page 9, line 28-31. Were the deaths found in these sources limited to those that occurred before 28 days of age?

Page 9, line 33: "The maternal admission data..... were examined to determine any maternal medical or pregnancy-related condition." What was done with this information e.g. were some extra stillbirth cases excluded? If so, how might this affect rates, as it reads as though these data were not also examined for cases without perinatal death?

Page 10, line 5: I suggest replacing the words "confirm a diagnosis" with "identify cases".

Page 10, line 10: could epidural analgesia for relief of pain during labour be differentiated from an epidural given to facilitate operative birth?

Page 10, line 19: there appears to be an error related to "non-mutually exclusive events" or perhaps the text needs further clarification.

Page 10, line 29: page 7 says births between July 2000 and June 2008 are included. Here it says January 2000 - December 2008.

Page 10, line 31: If the APDC includes 1.1 million admissions for these babies, it would suggest that, on average, around half of all babies have an admission after the birth one. Is it possible that admissions after 28 days were included in the output? If not, is there some other explanation for this very high rate.

Page 10, line 36 and page 11, line 28: "was significantly higher" is used but I could see no test of significance.

Page 10, line 38: please describe what the 'pre-specified medical reasons' are and specify how many women were excluded for this reason. Were the women in fact excluded, or the interventions not counted?

Page 10, line 38: How is an induction "for no medical reason" defined? Is gestation considered?

Page 11, line 5: severe perineal trauma does not appear to be defined.

Page 11, lines 30-39: There is no need to duplicate the data in the tables here.

Page 12, line 19: "circumcised immediately following the birth" might be replaced with "circumcised during the birth admission".

Page 12-13: It appears that the number of babies who were readmitted or those with diagnosis codes are used as the denominator. (I may be wrong about this, and some clarification would suffice.) If it is intended to say that babies born in one sector had a higher risk of readmission for (e.g.) infectious diseases, then all babies born in that sector need to be included in the denominator. This would seem to be more useful information than reporting "of all babies who were readmitted, infectious disease was a more common reason in those born in private than public hospitals" as it appears at present.

Page 12, line 27 should it read "reasons for transfer or readmission" rather than "reasons for readmission" as described in the methods section?

Page 13, line 33: also Watson et al.

Page 13, line 38: replace or qualify "as much as is possible" with "in this data set" or "in this detailed way".

Pages 14-17: can't be assessed until the text is revised after

	recalculation using all babies as the denominator. Tables 1 & 2 need to include p-values. Tables 3 & 4 need to specify in a footnote the variables used in adjusted analyses; reduce the number of decimal places in the confidence intervals to 1. Table 4: there appears to be an error in the number or percent reported for low Apgar scores. Tables 6 & 7 These require recalculation of all percentages and p-values using all babies as the denominator. I would like to see the paper again after the amendments have been made, particularly changing the denominator for the morbidity and transfer/readmission analyses. This is what is meant by 'requiring further specialist statistical review' above. I am not suggesting a need to find another reviewer.
--	---

REVIEWER	Myers, Jenny University of Manchester UK
REVIEW RETURNED	29-Dec-2013

GENERAL COMMENTS	This is a large retrospective analysis which has aimed to determine perinatal morbidity amongst low risk pregnancies. There is some very interesting and useful information contained within the paper but several presentation and analysis issues make it difficult to reach definitive conclusions. I have made several suggestions and comments below. Methods In general I think definition of several of the outcome measures could be improved. In the main, these have not been adequately defined in the methods section which makes interpretation of the results problematic. This is particularly important for the neonatal records which have been used in tables 5,6 & 7. It is not clear how these have been combined in table 7. For example feeding difficulties are different between sectors in table 5 but not table 6. Short term feeding difficulties clearly have different implications to longer term problems. Have they just been numerically combined? The same applies for intrauterine hypoxia, these methods need describing in more detail and the relative significance of the different codes needs to be considered.
--

Clearly, whilst many of the short term neonatal outcomes are important, justification for increased intervention in low risk women comes from a perceived reduction in the number of cases of HIE resulting in cerebral palsy and a reduction in perinatal mortality. Many parents and health care providers will accept an increased risk in short term complications if there is a demonstrated reduction in severe long term morbidity or death. I think the significant omission from the manuscript is an estimation of power to detect a difference in perinatal mortality. The authors state that there was no difference in perinatal mortality between the groups, however the CI for the OR are not far from being >1 in both groups. Did the authors not consider combining the groups (nullips and multips) to increase the power? Without a power calculation it is not possible to know whether the lack of effect reflects a type 2 error. Given the strength of the authors' conclusions, an estimation of power is essential in my view.

P1- Ln 35 If only births to 'low risk' women were included why was there need to further remove "all interventions for pre-specified medical reasons" as well. This is confusing and needs further clarification with details of numbers. Perhaps a flow diagram would help detailing the exclusions in steps. Data was analysed for less than half of the total nulliparous population. Whilst I appreciate the authors were aiming to analyse only low risk pregnancies, I think there would be extremely valuable information pertaining to the outcome of higher risk pregnancies between the two health care settings as well. Do the authors plan to report this in a separate paper? Why not include both groups in the same manuscript with a predefined subgroup analysis?

Where/how was the indication for induction recorded? Was this validated?

How was severe perineal trauma defined?

My points pertaining to the exclusion of higher risk pregnancies in the nulliparous group also apply to the multiparous group. In this section of the results P11, Ln 10-20, the authors use different terminology "exclusion for specific medical reasons", this is different to the paragraph relating to nulliparous pregnancies. I assume that the authors did not exclude CS for presumed fetal distress or failure to progress? This needs further clarification.

I assume the for the percentages quoted in paragraph 1, page 12, the denominator is the number of babies with a neonatal morbidity

record? Why was this used rather than the total number of births in each group. More babies in the public sector group had morbidity records, is this a difference in reporting or an increase in neonatal morbidity? Is the neonatal care also split by public and private sector?

How was "affected by forceps or vacuum delivery" defined? Does this imply scalp markings associated with instrumental delivery or more serious trauma? These percentages seem to reflect the increase in instrumental delivery in the private sector? Similarly, how was "intrauterine hypoxia" defined?

As a general point, where percentages are quoted in the text, p values are often not quoted. There are also no p values stated in tables 1 & 2. This is a significant omission and needs rectifying. It is not clear whether there are significant differences between the gestation groups in particular.

P14, Ln 35: the authors discuss the association between birth trauma and neonatal neurological morbidity. Were these outcomes not available in the current study. Proof that obstetric intervention was associated with an increase in these outcomes would provide much more credence to the authors' argument that obstetric intervention increases long term perinatal morbidity.

Were breastfeeding rates different between the two sectors? What evidence is there in the current study that epidural rates influence breastfeeding rates?

The authors comment that delivery at 37 weeks is associated with increased perinatal morbidity rates, in the current study the rates of delivery at this gestation are not different. More pregnancies were delivered at 38 weeks in the private sector group in the current study. I am not aware that there is evidence that deliveries after 38 weeks are also associated with an increase in neonatal morbidity? There is however an increase in perinatal morbidity in births after 40 weeks, this portion of the discussion needs more balance in my view.

I would have thought the abstract should include ORs and p values or similar?

P11 Ln 30 and 31 Typos in the AORs? Also rewrite "less likely to

	have no resuscitation", double negative which is confusing.
--	---

VERSION 1 – AUTHOR RESPONSE

Reviewer Name Mary-Ann Davey

This will be a really valuable contribution to the literature around public versus private maternity care.

Thank you

I have a number of questions/suggestions. Once these are addressed the responses to the questions above are all likely to be 'Yes'.

Title: Please change 'cohort' to 'retrospective cohort'

We are trying to use a similar description that we have used in the first private public paper published in the BMJ in 2012 and in all our other linked data studies. Linked data population based cohort studies are by their nature retrospective. We are happy to add the word 'retrospective' into the title if the Editor feels this is important but we are just concerned at the current length of the title and making the paper seem different to previous studies where we have used this title.

Page 6, line 10. Is it known who is and is not 'privately insured'? If they are not analysed by 'insurance' but by 'admission status', they should be referred to as "women admitted as private patients" or similar throughout the paper. In Australia, some women are privately insured, but choose not to use this insurance for maternity care (and are admitted as public patients), and others choose to be admitted as private patients despite not being insured. Most perinatal data collects the admission status, not the insurance status.

We have not used private insurance status as it is no longer possible in NSW to tell who was admitted to a public hospital with private insurance. This has been removed from the data set which is disappointing so we can only use public or private hospital admission. We have therefore, as we did in the first paper published in the BMJ in 2012, compared public and private hospital admissions. We realise however from your comment that this is not clear enough and we have detailed this more fully as we did in our first paper published in 2012. We have added the following to the methods for clarity.

"Hospitals are coded either as private or public in the data set. However, the data identifying women who received care in public hospitals under private accommodation status is no longer collected as it had been in the years 1996-97 and for this reason patients who are under private obstetric care in public hospitals are not able to be differentiated from their public counterparts, so for this study we analysed the data by hospital (private/public). A previous study published in 2000 (10) showed that there was a moderating factor on intervention rates when women with private insurance status gave birth in a public hospital, leading to lower intervention rates than when they gave birth in private hospitals."

Page 7, line 5: The paper states that 'these interventions were found to have increased by 5%...'. Perhaps provide a range as they are unlikely to have uniformly increased by 5%.

We have added the reference to the paper this came from [9] this was not data for this paper it was a single measure of change over a decade.

Page 7, line 22: Perhaps there is a word missing, but what is the 'New South Wales Centre'?

You are correct and we apologise. We have added the following

“New South Wales Centre for Health Record Linkage”

Page 7, line 34: Were re-admissions recorded in the APDC in July 2008 included to detect those in the first 28 days for births that occurred in June 2008?

The dataset included a cut off date for births 28 days prior to the cut off date for outcome assessment. This information was changed in the data sources section but has now also been added to the results section to indicate this.

Page 8, line 16: Please describe how cases below a certain linkage weight were managed in this study. If they were excluded, is there a risk that these would be more or less likely to be cases of mortality or morbidity?

There were 0.3% of cases with a low linkage rate obtained. This small number of cases (estimated to be <1000 in this low risk cohort) were excluded from the analyses. The cases with low linkage by definition are highly likely to have spurious outcomes. We have searched thorough other data linkage papers and can find no examples where the low weight linkage cases are included or analysed for comparison. We have also added the following to the limitations

“A small number of cases with a low linkage rate (0.3%) were not included and so there is the possibility of missing adverse outcomes. A previous study showed that where stillbirths are excluded due to low linkage these are at lower gestational ages and not term infants as were the focus in this study.”

48 Bentley JP, Ford JB, Taylor LK, Irvine KA, Roberts CL. Investigating linkage rates among probabilistic linked births and hospital records. BMC Medical Research Methodology. 12:149, 2012.

Page 8, line 20: Gestation is recorded more than one way - please specify which version was used in this study.

Gestation was recorded in completed weeks. The word ‘completed’ has been added to the description of the cohort in the methods section.

Page 8, line 24: It might help an international audience to describe the sorts of conditions that are (and are not) usually managed on the postnatal ward.

Thank you we have added the following to the Discussion

“While increasingly preterm babies >35 weeks/ and or >2.2k and some cases of jaundice may be managed at the bedside in some hospitals, this is less likely to occur in a private hospital. It is more likely to occur in large maternity units.”

Page 8, line 38 and Page 9, line 5: add the word 'singleton'.

Thanks for noting this we have now added the word 'singleton'.

Page 8, lines 7-11: this needs more explanation and justification. e.g. were babies born by caesarean section because of non-reassuring fetal status following a syntocinon infusion excluded? This might artificially reduce the apparent rate of induction, and the rate of CS etc. Which conditions collected from the APDC led to exclusion of cases? Were all women with these conditions excluded?

We have now clarified this by adding the word 'pre-existing medical reason' as they would not have been excluded if the caesarean section resulted from a non-reassuring heart rate following an augmented labour if the woman was otherwise low risk. Augmented labours were not excluded unless they were preceded by induced labours for medical reasons. The conditions excluded are detailed in the methodology. We have also added the following under outcomes

"If a caesarean section was undertaken during labour however for non-reassuring heart rate, dystocia etc these women were included in the study."

Page 9, line 28-31. Were the deaths found in these sources limited to those that occurred before 28 days of age?

Yes they were and they were counted only once. We have added the following
"Stillbirth and neonatal deaths were calculated from multiple sources but were limited to those that occurred 28 days from the birth and they were only counted once."

Page 9, line 33: "The maternal admission data..... were examined to determine any maternal medical or pregnancy-related condition." What was done with this information e.g. were some extra stillbirth cases excluded? If so, how might this affect rates, as it reads as though these data were not also examined for cases without perinatal death?

Yes, we found some additional deaths where the pre-existing maternal risk factors such as biliary atresia was recorded in other places than the MDC and so these were excluded. We believe this is one of the reasons that previous studies have shown a higher perinatal death rate in the public sector as women with more complex pregnancies tend to be cared for in the public sector. Not all pre-existing medical preconditions could be searched for due to the number of them and hence only the cases where death occurred was hand searching undertaken.

Page 10, line 5: I suggest replacing the words "confirm a diagnosis" with "identify cases".

This has now been changed

Page 10, line 10: could epidural analgesia for relief of pain during labour be differentiated from an epidural given to facilitate operative birth?

Unfortunately it cannot. We wish it could!

Page 10, line 19: there appears to be an error related to "non-mutually exclusive events" or perhaps the text needs further clarification.

The words "non-mutually exclusive" have been removed from the text. The analyses were only described this way to indicate that each infant could have one or more event occur and that each event was considered a unique incident.

Page 10, line 29: page 7 says births between July 2000 and June 2008 are included. Here it says January 2000 - December 2008.

This issue has been rectified and the text now indicates that the dates for inclusion are births between 1st July 2000 and 2nd June 2008 and outcome data 1st July 2000 and 30th Jun 2008.

Page 10, line 31: If the APDC includes 1.1 million admissions for these babies, it would suggest that, on average, around half of all babies have an admission after the birth one. Is it possible that admissions after 28 days were included in the output? If not, is there some other explanation for this very high rate.

This number refers to all admissions within the time frame as does the number of births – not just those which occurred within the first 28 days of life. We then limited our analyses to the low-risk cohort and then to those admissions which occurred within the first 28 days following birth. Even with this in mind, we agree that this rate is very high but a number of factors have to be taken into account. Firstly every admission to a SCN/NICU after the birth admission is a separate admission and this occurred in >10% of babies born to primiparous women even in this “low-risk cohort”. This number also includes all of the premature babies born within the time period as well as all of the transfers which occur. These are not however included in this paper

Page 10, line 36 and page 11, line 28: "was significantly higher" is used but I could see no test of significance.

Thank you for noting this we have changed it to “much higher.” Even though this would have been highly statistically significant

Page 10, line 38: please describe what the 'pre-specified medical reasons' are and specify how many women were excluded for this reason. Were the women in fact excluded, or the interventions not counted?

The pre-specified medical reasons were hypertension or diabetes (all that is available in the MDC) and then if an IOL or caesarean section before labour was recorded as occurring for a medical reason these were excluded from the data set. If the IOL or caesarean section had no medical reason listed it was included as it was considered highly likely done for a social reason and not a significant medical reason.

Page 10, line 38: How is an induction "for no medical reason" defined? Is gestation considered?

We excluded women who had babies under 37 or over 41 weeks so yes gestation was considered between those weeks. If the woman had no medical risk factors recorded (preeclampsia and/ diabetes) then she was also excluded, or if she smoked or the baby was under the 10th centile or over the 90th centile. If the women had a medical reason listed for the IOL she was also excluded but where no medical reason was listed this was included and considered to be likely to be an iatrogenic indicator of the model of care (Private/ Public)

Page 11, line 5: severe perineal trauma does not appear to be defined.

Thank you. We have now defined this as third and fourth degree perineal trauma.

Page 11, lines 30-39: There is no need to duplicate the data in the tables here.

We are happy to remove this if the Editor sees this as necessary. We are aware some people (consumers will read this as it is an open access journal) find tables difficult to understand and grasp meanings better in the written text.

Page 12, line 19: "circumcised immediately following the birth" might be replaced with "circumcised during the birth admission".

Thank you we agree with you and have changed this 'circumcised during the birth admission' as suggested.

Page 12-13: It appears that the number of babies who were readmitted or those with diagnosis codes are used as the denominator. (I may be wrong about this, and some clarification would suffice.) If it is intended to say that babies born in one sector had a higher risk of readmission for (e.g.) infectious diseases, then all babies born in that sector need to be included in the denominator. This would seem to be more useful information than reporting "of all babies who were readmitted, infectious disease was a more common reason in those born in private than public hospitals" as it appears at present.

It was never our intention to describe incidence. The numbers of uncommon events in a low risk population is too small to analyse by making the denominator all babies as many events had less than 10 cases and there were many more coding for rare conditions in the public sector due to the nature of a more complex clinical profile in this population. We consulted a biostatistician regarding this and as our aim is to determine morbidity for low risk women associated with high intervention rates during birth we only looked at the ICD 10 code for conditions which arise in the perinatal period (P00-P96) as described in the methods and identified in the tables. We took the number of admissions for those codes (P00-P96) as the denominator in order to determine differences that might exist due to the model of care, in particular the high rates of intervention and related perinatal morbidity.

Page 12, line 27 should it read "reasons for transfer or readmission" rather than "reasons for readmission" as described in the methods section?

Thank you we have changed this

Page 13, line 33: also Watson et al.

Sorry I do not know what this means? I can find no reference to Watson

Page 13, line 38: replace or qualify "as much as is possible" with "in this data set" or "in this detailed way".

Thank you. We have added "in this data set."

Pages 14-17: can't be assessed until the text is revised after recalculation using all babies as the denominator.

See explanation above

Tables 1 & 2 need to include p-values.

This is a follow up paper and we are following up the methodology for the previous paper in BMJ Open where we did not include p-values. We are happy to include them if the editor requires them. We are aiming for consistency with the first paper.

Tables 3 & 4 need to specify in a footnote the variables used in adjusted analyses; reduce the number of decimal places in the confidence intervals to 1.

The variables adjusted for are in the title of the paper and we consider this adequate.

We have been advised by the biostatistician who is an author on this paper that 3 significant figures should be the default provided sample size is sufficient (our sample is a large one). If the editor wants this changed we are happy to oblige

Table 4: there appears to be an error in the number or percent reported for low Apgar scores.

Thank you for identifying this error. We have now corrected this. But realized we made a mistake with the public Apgar scores as well which were the same and hence we have redone the OR and AOR. The Apgar score is no longer significantly different for multiparous women. We have added the following.

“While rates of Apgar scores of <7 at five minutes were slightly higher amongst primiparous women who gave birth in public hospital (but not multiparous women), and this has been demonstrated in another recent Australian publication [15], overall the babies were no more likely to be admitted to SCN/NICU compared to babies born in private hospitals.”

Tables 6 & 7 These require recalculation of all percentages and p-values using all babies as the denominator.

As discussed above

I would like to see the paper again after the amendments have been made, particularly changing the denominator for the morbidity and transfer/readmission analyses. This is what is meant by 'requiring further specialist statistical review' above. I am not suggesting a need to find another reviewer.

Reviewer Name J Myers

Institution and Country University of Manchester

UK

Please state any competing interests or state 'None declared': None declared

This is a large retrospective analysis which has aimed to determine perinatal morbidity amongst low risk pregnancies. There is some very interesting and useful information contained within the paper but several presentation and analysis issues make it difficult to reach definitive conclusions. I have made several suggestions and comments below.

Thank you for your comments. We realise the Australian context is slightly different to the UK and have aimed to explain this.

Methods

In general I think definition of several of the outcome measures could be improved. In the main, these have not been adequately defined in the methods section which makes interpretation of the results problematic.

This is particularly important for the neonatal records which have been used in tables 5,6 & 7. It is not clear how these have been combined in table 7. For example feeding difficulties are different between sectors in table 5 but not table 6.

Short term feeding difficulties clearly have different implications to longer term problems. Have they just been numerically combined? The same applies for intrauterine hypoxia, these methods need

describing in more detail and the relative significance of the different codes needs to be considered.

There are very small percentages in the readmission tables due to the fact this is a rare event when women are low risk and have term infants. These rates have been added together to get the outcome in Table 7. Due to the larger numbers in table 5 the results in table 7 will reflect this. There are 370 different codes in this section of the ICD10-AM system and grouping had to be undertaken for ease of analyses and reading. We have added more detail to the methods regarding these codes –

“Events were grouped in body systems where appropriate or under headings such as infection for ease of analysis and interpretation.”

Feeding problems of the newborn (P92) includes; P92.0 Vomiting in newborn, P92.2 Slow feeding in newborn, P92.3 Underfeeding of newborn, P92.4 Overfeeding of newborn, P92.5 Neonatal difficulty in feeding at breast, P92.8 Other feeding problems of newborn and P92.9 Feeding problems of newborn, unspecified. These codes do not provide for an analysis of the determinants of less compared to more serious feeding issues. The time period from birth perhaps could have assisted us in determining the degree of feeding difficulty but this type of work does not allow for such assumptions to be made. The same situation exists with the intrauterine hypoxia coding.

Clearly, whilst many of the short term neonatal outcomes are important, justification for increased intervention in low risk women comes from a perceived reduction in the number of cases of HIE resulting in cerebral palsy and a reduction in perinatal mortality. Many parents and health care providers will accept an increased risk in short term complications if there is a demonstrated reduction in severe long term morbidity or death. I think the significant omission from the manuscript is an estimation of power to detect a difference in perinatal mortality. The authors state that there was no difference in perinatal mortality between the groups, however the CI for the OR are not far from being >1 in both groups. Did the authors not consider combining the groups (nullips and multips) to increase the power? Without a power calculation it is not possible to know whether the lack of effect reflects a type 2 error. Given the strength of the authors' conclusions, an estimation of power is essential in my view.

We have consulted with a biostatistician regarding this and it is not considered appropriate to calculate power in retrospective population data sets. We have looked at the perinatal mortality for primip and multip groups combined and there was no statistical difference but we were advised this may mask a higher perinatal death in primips and so we judged it prudent to be transparent and separate the two groups out and again this was not significant. The chance of a Type 2 error is limited due to the size of the data set. Using the smallest sample size we have in this study (28703 – Private/multip cohort) gives this study the power to detect a difference in total perinatal mortality from 8/1000 to 8.75/1000 (90% power, alpha 0.05). If the editor wants this included in the methods we are happy to do so but inclusion of power in retrospective population data sets is not considered appropriate.

P1- Ln 35 If only births to 'low risk' women were included why was there need to further remove “all interventions for pre-specified medical reasons” as well. This is confusing and needs further clarification with details of numbers. Perhaps a flow diagram would help detailing the exclusions in steps. Data was analysed for less than half of the total nulliparous population. Whilst I appreciate the authors were aiming to analyse only low risk pregnancies, I think there would be extremely valuable information pertaining to the outcome of higher risk pregnancies between the two health care settings as well. Do the authors plan to report this in a separate paper? Why not include both groups in the same manuscript with a predefined subgroup analysis?

We agree this would be interesting way to look at the data and could be done separately. In Australia

the profile of patients is very different in private and public sectors. The women in the public sector are more socially disadvantaged and have more medical complications so our aim it to try and match them without this added risk in the public sector masking true outcomes. The low risk cohort was determined by the authors based on our previous paper published in 2012 in BMJ Open using the same population and definitions of risk. In that paper we modeled the method on a paper done 10 years earlier to show in the same low risk population over a decade the rates of intervention had increased. This paper is following this data set up for neonatal outcomes.

Where/how was the indication for induction recorded? Was this validated?

Indication for IOL was as recorded on the perinatal data collection. This was not validated in this study as this has been undertaken previously by another Australian research group. Lain, S, Hadfield, R, Raynes-Greenow C; Ford J; Mealing N; Algert C; Roberts C. Quality of Data in Perinatal Population Health Databases: A Systematic Review. Medical Care 2012;50;4; e7–e20 reviewed five studies and found a kappa of >0.75 between data collections made at birth (such as the PDC) and hospital records.

How was severe perineal trauma defined?

We have added in brackets that this includes 3rd and 4th degree perineal trauma.

My points pertaining to the exclusion of higher risk pregnancies in the nulliparous group also apply to the multiparous group. In this section of the results P11, In 10-20, the authors use different terminology “exclusion for specific medical reasons”, this is different to the paragraph relating to nulliparous pregnancies. I assume that the authors did not exclude CS for presumed fetal distress or failure to progress? This needs further clarification.

Any women who is low risk when labour starts and has not been induced or had a booked caesarean for an identified medical reason was included. This means if fetal distress develops in labour or there is failure to progress they are included in this data set. We have now clarified this in the text. With the following statement

“If a caesarean section was undertaken during labour however for non-reassuring heart rate, dystocia etc these women were included in the study.”

I assume the for the percentages quoted in paragraph 1, page 12, the denominator is the number of babies with a neonatal morbidity record? Why was this used rather than the total number of births in each group. More babies in the public sector group had morbidity records, is this a difference in reporting or an increase in neonatal morbidity?

It was never our intention to describe incidence. The numbers of uncommon events in a low risk population is too small to analyse by making the denominator all babies as many events had less than 10 cases and there were many more coding for rare conditions in the public sector due to the nature of a more complex clinical profile in this population. We consulted a biostatistician regarding this and as our aim is to determine morbidity for low risk women associated with high intervention rates during birth we only looked at the ICD 10 code for conditions which arise in the perinatal period (P00-P96). We took the number of admissions for those codes (P00-P96) as the denominator in order to determine differences that might exist due to the model of care, in particular the high rates of intervention and related perinatal morbidity

Is the neonatal care also split by public and private sector?

Yes it is. Even if the baby is transferred to a public hospital or readmitted to a public hospital the place of birth is still considered to be where the baby was actually born (private or public hospital)

How was “affected by forceps or vacuum delivery” defined? Does this imply scalp markings associated with instrumental delivery or more serious trauma? These percentages seem to reflect the increase in instrumental delivery in the private sector?

Yes we agree this does reflect the higher rate of instrumental birth which is the purpose of this paper. We wanted to see if the higher rates of neonatal morbidity were seen when higher rates of medical intervention was done on low risk women. This was a combined groupings of the ICD-10-AM codes P10-P15 which cover all forms of birth trauma. This is broken down more fully in the figure.

Similarly, how was “intrauterine hypoxia” defined?

Please see answer above

As a general point, where percentages are quoted in the text, p values are often not quoted. There are also no p values stated in tables 1 & 2. This is a significant omission and needs rectifying. It is not clear whether there are significant differences between the gestation groups in particular.

All the gestational groups are different which is why we adjusted for this. This is a follow up paper and we are following up the methodology for the previous paper in BMJ Open where we did not include p-values. We are happy to include them if the editor requires them.

P14, Ln 35: the authors discuss the association between birth trauma and neonatal neurological morbidity. Were these outcomes not available in the current study. Proof that obstetric intervention was associated with an increase in these outcomes would provide much more credence to the authors’ argument that obstetric intervention increases long term perinatal morbidity.

We agree with your comments and our ongoing work is now looking more directly at longer-term outcomes in these babies but this will be a separate paper

Were breastfeeding rates different between the two sectors?

Unfortunately breastfeeding is only now recorded in data collected and was not available for the data set we analysed. We hope to be able to do this in future. Only feeding difficulties are recorded in the APDC. To have this documented associated with birth or on readmission means they are quite significant.

What evidence is there in the current study that epidural rates influence breastfeeding rates?

There is no evidence that epidural rates influence breastfeeding rates in this study and as we do not have method of infant feeding recorded this makes it difficult to look for associations. We are simply in the discussion reviewing the literature and commenting on possible associations shown in the literature.

The authors comment that delivery at 37 weeks is associated with increased perinatal morbidity rates, in the current study the rates of delivery at this gestation are not different. More pregnancies were delivered at 38 weeks in the private sector group in the current study. I am not aware that there is evidence that deliveries after 38 weeks are also associated with an increase in neonatal morbidity? There is however an increase in perinatal morbidity in births after 40 weeks, this portion of the

discussion needs more balance in my view.

There are several large publications showing a relationship with early elective delivery and increased morbidity and we have cited some of these. We agree that we have not mentioned the increased perinatal morbidity associated with going over 40 weeks and we have now included this to provide the balance you right indicate is missing.

“However there is also evidence that there are increased adverse perinatal outcomes for babies born following 41 completed weeks but we did not examine this population [49].”

49. Gülmezoglu AM, Crowther CA, Middleton P, Heatley E. Induction of labour for improving birth outcomes for women at or beyond term. CochraneDatabase of Systematic Reviews 2012;Issue 6. Art.No.:CD004945. DOI: 10.1002/14651858.CD004945.pub3.

I would have thought the abstract should include ORs and p values or similar?

We are happy to add these in but were worried it would make the abstract too long. We will add them if the Editor advises this.

P11 Ln 30 and 31 Typos in the AORs?

I am sorry we could not find the typos you refer to

Also rewrite “less likely to have no resuscitation”, double negative which is confusing.

We have changed this to:

“or not to be resuscitated.”

VERSION 2 – REVIEW

REVIEWER	Mary-Ann Davey Judith Lumley Centre, La Trobe University Australia
REVIEW RETURNED	12-Mar-2014

GENERAL COMMENTS	This paper will be a valuable contribution to the literature regarding elective interventions in labour and birth once a few problems have been corrected. The main flaw in this paper is the way in which morbidity has been calculated and reported. The proportion of babies born in the public and private sector with specified morbidity is compared in numerous places. However the denominators used are the number of babies with a ‘morbidity record’ (what does this mean?) in the birth episode, or the number transferred or readmitted for the period after the birth episode. The denominator should be the number of babies born to low-risk women in the sector. This would allow meaningful comparisons between the sectors which is not possible at present. This paper will be a valuable contribution to the literature regarding elective interventions in labour and birth once a few problems have been corrected. The main flaw in this paper is the way in which morbidity has been calculated and reported. The proportion of babies born in the public
--

and private sector with specified morbidity is compared in numerous places. However the denominators used are the number of babies with a 'morbidity record' (what does this mean?) in the birth episode, or the number transferred or readmitted for the period after the birth episode. The denominator should be the number of babies born to low-risk women in the sector. This would allow meaningful comparisons between the sectors which is not possible at present.

I will list my other comments/questions in order through the paper.

Abstract: If the design is said to be a cohort study it should be specified as a retrospective cohort study.

-The number of participants should be the number of low-risk women studied, as I don't believe any of the other 600,000 are considered here.

-The interventions are not mentioned as outcomes.

-The statement about admission for the conditions being more likely in the private sector needs to be revised once the denominator issue is corrected. (It will probably still be true of some or all of the conditions.) (This applies where veer comparisons are made regarding any morbidity in the paper.)

Key messages: 2nd needs to be reconsidered.

Strengths and limitations: change the 'over half a million women' comment (and I will ask later about the 1.1million admissions).

Introduction:

-The women should not be referred to as privately insured as this information is not known. They are admitted as private patients who are mostly (but not always) insured, while the public women are admitted as public patients who may or may not be privately insured. Please correct throughout.

-2.2% of women appear not to be accounted for (32.8+65%).

Data sources:

-2nd para – sound a little like the dataset has only 1/3 of births. Might be better to describe as 'The NSW PDC contains statistics on all births in NSW which make up...'. The APDC is said to be used to 'further exclude women' when exclusions have not yet been mentioned. More details of the linkage -weightings that were accepted would be helpful.

Gestation – 2 variables are mentioned. Which was used (at birth, or mentstrual)?

Subjects:

-Does a pre-existing medical condition include a complication of

	pregnancy? Outcomes: -2nd para- The meaning of the sentence beginning “The maternal admission...” is not clear to me. The information about the primary cause of death does not appear to be used in the analyses.-Is there a reason why augmentation of labour was not considered? Results: -The meaning of the 1.1 million admissions is not clear. Does this mean 1 admission for each birth and an average of one extra admission for every 2 births? If so, it seems very high.-Might it not be better to use a word other than ‘significantly’ when reporting results for which no tests of significance are reported? Perinatal characteristics...: -The birthweights are not shown in the tables. Reason for birth admission...: -Please explain what is meant by a ‘morbidity record’ and which cases would be expected to have one. Is there any financial (or other) incentive for one sector more than the other to have one?-Is the ‘complexity’ mentioned referring to clinical complexity, or coding/recording complexity?-All comparisons of morbidities between the public and private sectors need to include all babies born to low risk women in the denominator, not just those who were admitted. Discussion: -Intervention rates – last sentence ‘... nto statistically different between babies born in public and private hospitals when..’. Neonatal admission.. change denominators. -Please provide reference for Hyperbilirubinaemia paper, and put this sentence into context.-Para beginning ‘Twice as many...’ It might help to have a table of the number of babies affected by the various conditions in birth admission and/or readmission and report this as a % of all births.-The sentence about corticoneurogenesis might be clarified by adding the words ‘by early deliver’ at the end. Conclusion: -The final sentence is not supported by the data as I don’t believe
--	---

	outcomes were examined by gestation or birth type. Ref 49 'spending' is misspelled. Fig 1 & Tables titles need to specify 'low-risk'. Tables 3 & 4 – suggest reducing number of decimal places in the CIs Tables 5 & 6 need denominator corrected and %s recalculated Table 6 respiratory disease in privates appears incorrect. Final 3 figures have no title or number in my version. They need to be amended with new denominators.
--	---

VERSION 2 – AUTHOR RESPONSE

This paper will be a valuable contribution to the literature regarding elective interventions in labour and birth once a few problems have been corrected.

Thanks for your encouragement and input into this paper.

The main flaw in this paper is the way in which morbidity has been calculated and reported. The proportion of babies born in the public and private sector with specified morbidity is compared in numerous places. However the denominators used are the number of babies with a 'morbidity record' (what does this mean?) in the birth episode, or the number transferred or readmitted for the period after the birth episode. The denominator should be the number of babies born to low-risk women in the sector. This would allow meaningful comparisons between the sectors which is not possible at present.

We have obviously not explained this adequately and we agree it is confusing. While we consider the way we have done it to be entirely acceptable and as stated previously we have had biostatistical advice on it we feel that it is a major sticking point for you and so have changed it and undertaken the substantial work needed as a result of this change.

I will list my other comments/questions in order through the paper.

Abstract: If the design is said to be a cohort study it should be specified as a retrospective cohort study.

As stated before we are trying to use a similar description that we have used in the first private public paper published in the BMJ in 2012 and in all our other linked data studies. Linked data population based cohort studies are by their nature retrospective. We are happy however to add the word 'retrospective' into the abstract as requested.

-The number of participants should be the number of low-risk women studied, as I don't believe any of

the other 600,000 are considered here.

It is standard practice to describe the overall population from where the data is sourced and how you obtained the population for study. This is how we and indeed all authors present data for studies like this. We make clear the numbers of low risk women we end up with. We are following exactly the same format as the first paper published in this journal. If the editor thinks this needs to be changed we will comply

-The interventions are not mentioned as outcomes.

The interventions are not an outcome as this was reported in the first paper. The outcomes are the perinatal outcomes associated with private or public hospital birth. We hope this is now clear.

-The statement about admission for the conditions being more likely in the private sector needs to be revised once the denominator issue is corrected. (It will probably still be true of some or all of the conditions.) (This applies where veer comparisons are made regarding any morbidity in the paper.)

See above comments

Key messages: 2nd needs to be reconsidered.

Strengths and limitations: change the 'over half a million women' comment (and I will ask later about the 1.1million admissions).

We have changed this to a large data set instead of a number

Introduction:

-The women should not be referred to as privately insured as this information is not known. They are admitted as private patients who are mostly (but not always) insured, while the public women are admitted as public patients who may or may not be privately insured. Please correct throughout.

As stated before we have not used private insurance status as it is no longer possible in NSW to tell who was admitted to a public hospital with private insurance. This has been removed from the data set which is disappointing so we can only use public or private hospital admission. We have therefore, as we did in the first paper published in the BMJ in 2012, compared public and private hospital admissions. We realise however from your comment that this is not clear enough and we have detailed this more fully as we did in our first paper published in 2012. We have added the following to the methods for clarity. We have checked that privately insured is not in the paper unless in the introduction reporting on other studies. We cannot find this other than in the introduction which is appropriate and when discussing other studies or privately insured women in general.

-2.2% of women appear not to be accounted for (32.8+65%).

We have now updated this to the latest data and inserted the following:

In Australia, the national statistics reveal that women who gave birth in hospital 29% (n =83,573) gave birth in private hospitals directly under private obstetric care [1]. The remaining 71% (n=204,399) of women gave birth in public hospitals in Australia. Women who are privately insured have been reported to have better maternal and perinatal outcomes compared to women who give birth in public hospitals as public patients [2]

Data sources:

-2nd para – sound a little like the dataset has only 1/3 of births. Might be better to describe as ‘The NSW PDC contains statistics on all births in NSW which make up....’.

We have added in brackets that this is all births in NSW which we thought was clear from the way we had written it.

The APDC is said to be used to ‘further exclude women’ when exclusions have not yet been mentioned. More details of the linkage -weightings that were accepted would be helpful.

This is simply a double check to make sure we excluded any women with risk factors it has nothing to do with linkage weighting and we have already explained how this is done. See below from our previous response.

There were 0.3% of cases with a low linkage rate obtained. This small number of cases (estimated to be <1000 in this low risk cohort) were excluded from the analyses. The cases with low linkage by definition are highly likely to have spurious outcomes. We have searched thorough other data linkage papers and can find no examples where the low weight linkage cases are included or analysed for comparison. We have also added the following to the limitations

“A small number of cases with a low linkage rate (0.3%) were not included and so there is the possibility of missing adverse outcomes. A previous study showed that where stillbirths are excluded due to low linkage these are at lower gestational ages and not term infants as were the focus in this study.”

Gestation – 2 variables are mentioned. Which was used (at birth, or menstrual)?

All gestational ages now are a combination of LMP and adjusted with early ultrasound if a difference is noted. Therefore this date will be used to calculate the gestational age at birth and this is what is used as in every maternity unit in Australia.

This is stated in the paper already on page 6:

Gestation is recorded at birth and is also recorded in the database according to the woman's menstrual history, usually combined with a routine scan at 12-13 weeks.

Subjects:

-Does a pre-existing medical condition include a complication of pregnancy?

Yes, we have excluded pre-existing medical complications and complications in pregnancy as state before.

Outcomes:

-2nd para- The meaning of the sentence beginning ‘The maternal admission...’ is not clear to me. The information about the primary cause of death does not appear to be used in the analyses.

We apologise as we initially included reason for death but it made the paper too dense and added nothing further. We have now removed these following couple of sentences.

Reasons for stillbirth and neonatal death were taken from the principal cause noted on the death registration. If the death was not yet registered, the principal diagnosis as recorded in the neonate's birth or subsequent admission was utilised. In any case where either of these two methods did not supply a reason for death or principal diagnosis, no reason for death was recorded.

-Is there a reason why augmentation of labour was not considered?

We have not considered this has been shown to be unreliable in this data set.

Results:

-The meaning of the 1.1 million admissions is not clear. Does this mean 1 admission for each birth and an average of one extra admission for every 2 births? If so, it seems very high.

As stated before in response to this question: This number refers to all admissions within the time frame as does the number of births – not just those which occurred within the first 28 days of life. We then limited our analyses to the low-risk cohort and then to those admissions which occurred within the first 28 days following birth. Even with this in mind, we agree that this rate is very high but a number of factors have to be taken into account. Firstly every admission to a SCN/NICU after the birth admission is a separate admission and this occurred in >10% of babies born to primiparous women even in this “low-risk cohort”. This number also includes all of the premature babies born within the time period as well as all of the transfers which occur. These are not however included in this paper. We realise the addition of 28 days was incorrect and we have also made clear that for this time period of eight years some of the admission could be for children

We have clarified this further by changing the following sentence to:

The PDC dataset for the time period July 1st 2000 to 2nd June 2008 contained the antenatal, birth and postnatal details on 691 738 births. The APDC for the time period July 1st 2000 to 30th June 2008 contained >1.1 million admissions for the neonates/children of these women.

-Might it not be better to use a word other than ‘significantly’ when reporting results for which no tests of significance are reported?

We have deleted this

Perinatal characteristics...:

-The birthweights are not shown in the tables.

The only babies included were in the 10th to 90th centile, so within normal birth weights, so not included in the tables. We unclear as to what this would add?

Reason for birth admission...:

-Please explain what is meant by a ‘morbidity record’ and which cases would be expected to have one. Is there any financial (or other) incentive for one sector more than the other to have one?

We have added ‘ed’ on the end as this was missing (morbidity recorded). All cases with a morbidity would be recorded. We have changed the sentence to make it clear:

We examined neonatal morbidity as coded on the neonatal birth admission record and found fewer babies overall had a morbidity recorded (ICD-10-AM code other than the birth code)

Both sectors would have a financial incentive to record all the admissions and we are reluctant to start accusing either the private or public sector without evidence.

-Is the ‘complexity’ mentioned referring to clinical complexity, or coding/recording complexity?

We have changed this to clinical and social maternal complexity

-All comparisons of morbidities between the public and private sectors need to include all babies born to low risk women in the denominator, not just those who were admitted.

See above comments

Discussion:

-Intervention rates – last sentence ‘... nto statistically different between babies born in public and private hospitals when..’.

We do not know what is wrong with this sentence.

Neonatal admission.. change denominators.

See above

-Please provide reference for Hyperbilirubinaemia paper, and put this sentence into context.

We have added the references and clarified this

-Para beginning ‘Twice as many...’ It might help to have a table of the number of babies affected by the various conditions in birth admission and/or readmission and report this as a % of all births.

Please see table 7

-The sentence about corticoneurogenesis might be clarified by adding the words ‘by early deliver’ at the end.

This has been changed to :

Some claim that during the final weeks of gestation the fetal brain goes through a marked increase in mass and nerve growth (corticoneurogenesis) which may be best left undisturbed by allowing the normal gestational length to occur [42].

Conclusion:

-The final sentence is not supported by the data as I don’t believe outcomes were examined by gestation or birth type.

We have been cautious about this as we have said ‘may be’.

Early term delivery and instrumental births may be associated with increased morbidity in neonates and this requires urgent attention.

Ref 49 ‘spending’ is misspelled.

There is no ‘spending’ in reference 49 but have corrected an error in reference 47

Fig 1 & Tables titles need to specify ‘low-risk’.

This has been added

Tables 3 & 4 – suggest reducing number of decimal places in the CIs Tables 5 & 6 need denominator corrected and %s recalculated

We have changed this

Table 6 respiratory disease in privates appears incorrect.

This has changed over all due to the change in the denominator

As stated in the methodology

Final 3 figures have no title or number in my version. They need to be amended with new denominators.

There are only two figures and both have titles?

Please note we have calculated the circumcision rates out of male infants only as is reported national and previously this was out of all babies.

VERSION 3 - REVIEW

REVIEWER	Mary-Ann Davey Judith Centre, La Trobe University Australia
REVIEW RETURNED	20-Apr-2014

GENERAL COMMENTS	Thank you for addressing some of the major issues. This has now enabled a clearer review of the paper which has unfortunately resulted in some new matters becoming apparent at this late stage. Most can be easily addressed. The unchecked boxes above will then all be able to become 'yes' , especially the accuracy of the abstract (once overall perinatal mortality is reported and the conclusion is amended), and the conclusion being justified by the results. This paper is now much more informative. Since perinatal mortality and morbidity are the outcomes of interest, mortality should be reported for all babies (as well as for primiparous and multiparous women separately) in the same way that morbidity is reported. The conclusion in the abstract implies that outcomes were analysed for women who had and did not have intervention. It might be more accurately expressed as "For low risk women, care in a private hospital which includes higher rates of intervention appears to be associated with higher rates of morbidity seen in the neonate, and no evidence of a reduction in perinatal mortality" (if this remains so once overall mortality is reported). A reference should be provided for the statement in the 2nd paragraph that the perinatal mortality rate is not declining . (The Victorian CCOPMM report would be one). Lines 14-16 on page 11 seem not to have been updated to describe the new denominator. Line 7 on page 12 should read ".. babies born to low risk women in a public or private hospital...". Please provide a p-value for Figure 1. Lines 7-9 on page 14 state that differences were "due to the added complexity..". This is a possible explanation I think that belongs in the discussion rather than a result.
--

	Does Figure 2 report results for all babies rather than only for those born to low-risk women? If so, this should be made clear and justified. Line 35 on page 16 appears to be missing the words "more common" after "significantly". Line 18 on page 17 appears to report an analysis that is not in the paper i.e. the relationship between gestation and elective delivery in the public and private sectors. The addition of the word "possibly" before "due to" would be better. I think the first word on line 33 on page 19 should be "weight" rather than "rate". There is no footnote for the asterisk in Table 2. The reference categories for the Apgar score Odds Ratio in Table 4 appear to have been reversed (using public as the reference). The conclusion in the body of the paper focuses on the relationship between interventions and morbidity and mortality, while the analyses in the paper focussed on the relationship between private or public hospital and mortality and morbidity. This needs modifying.
--	--

VERSION 3 – AUTHOR RESPONSE

Thank you for addressing some of the major issues. This has now enabled a clearer review of the paper which has unfortunately resulted in some new matters becoming apparent at this late stage. Most can be easily addressed.

The unchecked boxes above will then all be able to become 'yes', especially the accuracy of the abstract (once overall perinatal mortality is reported and the conclusion is amended), and the conclusion being justified by the results.

This paper is now much more informative.

Since perinatal mortality and morbidity are the outcomes of interest, mortality should be reported for all babies (as well as for primiparous and multiparous women separately) in the same way that morbidity is reported.

We are happy to provide the overall mortality rate which is not significantly different with the unadjusted OR being (1.35; CI 0.95-1.91) and AOR being (1.37; CI 0.96-1.97). We have had a biostatistician involved with this paper who felt as we did that it was much more transparent to separate the groups into primip and multip to be clear we were not hiding a potential increased mortality in primips under the combined groups. As with the UK Place of Birth study we are trying to be very careful to look at these groups of women separately as the perinatal mortality rate is higher in primips generally. I do not see the benefits of an additional combined table as there are already so many tables and figures in this paper. We have included the combined rate so you can see there is no attempt to hide poor figures in fact we have been very careful to show we are in no way trying to hide adverse outcomes

The conclusion in the abstract implies that outcomes were analysed for women who had and did not have intervention. It might be more accurately expressed as "For low risk women, care in a private

hospital which includes higher rates of intervention appears to be associated with higher rates of morbidity seen in the neonate, and no evidence of a reduction in perinatal mortality" (if this remains so once overall mortality is reported).

We are happy to change this

A reference should be provided for the statement in the 2nd paragraph that the perinatal mortality rate is not declining . (The Victorian CCOPMM report would be one).

We will add the latest national report

Lines 14-16 on page 11 seem not to have been updated to describe the new denominator. We have updated this and to make it really clear we have added total before number: The TOTAL number of babies born in a public or a private hospital were used as the denominator when calculating the percentage of babies born with a morbidity code attached to their birth record or the number of babies readmitted with a designated morbidity code. This methodology provides for comparison between place of birth taking into consideration the fact that up to 55 morbidity codes can be attached to any one birth or readmission record

Line 7 on page 12 should read ".. babies born to low risk women in a public or private hospital...".

We have made it clear that the low risk women were both on the private and public hospital now

Please provide a p-value for Figure 1.

This seems a very odd request and putting p values against each column in the figure is very unusual. The p value is available in table seven and this is adequate. The purpose of the figure is to show where the difference in trauma lies, which is scalp trauma.

Lines 7-9 on page 14 state that differences were "due to the added complexity..". This is a possible explanation I think that belongs in the discussion rather than a result.

We think there needs to be some explanation here and we have also addressed this in the discussion.

Does Figure 2 report results for all babies rather than only for those born to low-risk women? If so, this should be made clear and justified.

We have added to Figure 2 title: "Birth trauma as a percentage of all births to low risk women in private and public hospitals" to make this clear.

Line 35 on page 16 appears to be missing the words "more common" after "significantly".

Thank you we have added this

Line 18 on page 17 appears to report an analysis that is not in the paper i.e. the relationship between gestation and elective delivery in the public and private sectors. The addition of the word "possibly" before "due to" would be better.

Thanks we have added the word 'possibly' before 'due to'

I think the first word on line 33 on page 19 should be "weight" rather than "rate". This is the

terminology that CHeReL report . We have added (false/positive) now exactly as used by CHeReL.

A small number of cases with a low linkage (false/positive) rate (0.3%) were not included and so there is the possibility of missing adverse outcomes. A previous study showed that where stillbirths are excluded due to low linkage these are at lower gestational ages and not term infants as were the focus in this study [45]

There is no footnote for the asterisk in Table 2.

Sorry we have removed this.

The reference categories for the Apgar score Odds Ratio in Table 4 appear to have been reversed (using public as the reference).

Thank you very much for picking this we have rechecked all the data and you are correct the numbers were the wrong and we had missed a digit off one. We have now made clear that the incidence of Apgar scores of less than 7 at 5 minutes is higher in the public sector. We have changed as below:

Babies of primiparous women who gave birth in a private hospital were less likely to have an Apgar of <7 at five minutes (AOR 1.34 95% CI 1.18-1.53; $p < 0.001$) as were babies of multiparous women who gave birth in private hospitals (AOR 1.37 95% CI 1.14-1.64). babies born in private hospitals were less likely

Rates of Apgar scores of <7 at five minutes were slightly higher amongst low risk women who gave birth in public hospital, and this has been demonstrated in another recent Australian publication [15], overall the babies were no more likely to be admitted to SCN/NICU compared to babies born in private hospitals.

The conclusion in the body of the paper focuses on the relationship between interventions and morbidity and mortality, while the analyses in the paper focussed on the relationship between private or public hospital and mortality and morbidity. This needs modifying.

We have added one more sentence and inserted especially in private hospitals in the conclusion:

Previous claims that high intervention rates in private hospitals lead to better perinatal outcomes than those seen in public hospitals need to be questioned.